# ROYAL SOCIETY
# OPEN SCIENCE

behaviour/cognition/evolution

social learning, cultural evolution, computational modelling, collective behaviour, decision-making

**Author for correspondence:**
Dominik Deffner
e-mail: dominik_deffner@eva.mpg.de

# Dynamic social learning in temporally and spatially variable environments

## Dominik Deffner, Vivien Kleinow and Richard McElreath

Max Planck Institute for Evolutionary Anthropology, Department of Human Behavior, Ecology and Culture, Leipzig, Germany

DD, 0000-0002-1649-3861; RM, 0000-0002-0387-5377

Cultural evolution is partly driven by the strategies individuals use to learn behaviour from others. Previous experiments on strategic learning let groups of participants engage in repeated rounds of a learning task and analysed how choices are affected by individual payoffs and the choices of group members. While groups in such experiments are fixed, natural populations are dynamic, characterized by overlapping generations, frequent migrations and different levels of experience. We present a preregistered laboratory experiment with 237 mostly German participants including migration, differences in expertise and both spatial and temporal variation in optimal behaviour. We used simulation and multi-level computational learning models including time-varying parameters to investigate adaptive time dynamics in learning. Confirming theoretical predictions, individuals relied more on (conformist) social learning after spatial compared with temporal changes. After both types of change, they biased decisions towards more experienced group members. While rates of social learning rapidly declined in rounds following migration, individuals remained conformist to group-typical behaviour. These learning dynamics can be explained as adaptive responses to different informational environments. Summarizing, we provide empirical insights and introduce modelling tools that hopefully can be applied to dynamic social learning in other systems.

## 1. Introduction

Humans are highly invasive apes. Starting from African origins, our ancestors have populated and successfully made a living in virtually any habitable region around the globe. This immensely flexible behavioural adaptation relies on the population-level accumulation and modification of cultural information over time that results in the emergence and refinement of locally adapted tools, beliefs and institutions [1–3]. Such behavioural or cultural evolution is partly driven by the strategies individuals employ

to learn behaviour from others, i.e. the rules that govern how information is passed on between interacting individuals [4–6]. Both formal models and controlled experiments have established how humans (should) combine individual and social information strategically to acquire locally adaptive information (e.g. [6,7], for reviews of the theoretical end empirical literature, respectively). Previous laboratory experiments let fixed groups of individuals engage in repeated rounds of a learning task and analysed how individuals' choices are affected by the payoffs they received and the choices of other members of their group. Typical findings are that social learning is generally adaptive but underused [8,9], individuals use a combination of payoff-biased, frequency-dependent and a set of other strategies [9,10], social learning strategies can regulate the 'wisdom/madness' of collective decision making [11] and, finally, there are considerable and consistent inter-individual differences in the reliance on social information [12,13].

Our study includes three features that are critical in natural populations but were missing from previous experimental approaches: (i) differences in expertise among group members, (ii) the presence of both temporal and spatial variation in optimal behaviour, and (iii) analysis of time dynamics in learning. To better understand the adaptive logic of culture in real organisms, we must not only study learning processes in isolation but investigate how learning intersects with demography and operates in dynamic groups [14].

Group membership in previous experiments was constant and all participants had the same level of experience in their current environment. Understanding the strategic learning decisions individuals make under such circumstances is elucidating, but those experiments do not reflect decision making in the real world. Natural populations are characterized by age structure, overlapping generations and frequent migrations between different habitats. Juveniles, for instance, grow up in an informational environment where they can learn from social interaction partners of different ages and, thereby, different levels of experience. In the social learning literature, organisms as different as young guppies and human children have been proposed to follow a 'copy older over younger models' strategy [15,16]. Whether copying older rather than younger individuals is adaptive, however, is not straightforward, but depends on the relative strength of different interacting forces. These forces include the importance of a cultural trait for survival, the difficulty to acquire the trait as a juvenile and the rate of environmental change [17].

Birth and death are also not the only processes that result in such an experience gradient among demonstrators. Migration between different habitats similarly results in an experience-structured population where group members have had different numbers of learning opportunities in the present environment. Formal modelling has explored the consequences of such spatial variability on the evolution of learning as compared with temporal changes in the environment that occur to everyone at the same time [18]. Under pure spatial variation, environmental factors vary across a spatial transect, i.e. from one habitat to the next, but are constant over time. Under pure temporal variation, by contrast, factors vary over time but are constant across space [7,19–21]. Most natural environments vary in both space and time, and adaptive learning strategies can be expected to differ depending on the dominant mode of environmental variation. An important finding, for instance, is that conformist learning is generally adaptive in spatially varying environments but not so much in temporally varying environments. Conformity helps migrants to efficiently adopt community-typical behaviour and also lets residents filter out the non-adaptive variation brought in by migrating individuals. Both factors tend to increase proportions of adaptive behaviour in the population (especially if individuals must choose between more than two traits) [18]. If the environment changes temporally, by contrast, everyone becomes non-adapted at the same time, such that conformity and social learning in general are expected to be of less value. Despite its theoretical relevance, this difference in adaptive social learning strategies between spatially and temporally varying environments, has received little empirical attention.

Formal models have also explored the effects of migration and conformist cultural transmission on within- and between-group cultural diversity [4,22]. General findings are that conformist learning tends to increase and high migration rates tend to decrease between-group cultural diversity (the opposite associations are true for diversity within groups). These models further highlight the need for more empirical research into the individual-level learning strategies that underlie acculturation in spatially varying environments [22].

In this paper, we report a laboratory social learning group experiment that investigates the dynamics of strategic social learning in experience-structured groups with both spatial and temporal variability (see preregistration at https://osf.io/a5bkg/). Such 'microsociety' experiments create social contexts in which groups of individuals can evolve behavioural traditions, through a combination of individual exploration of the environment and the available social information [23,24]. Unlike many other experimental studies,

social information in these experiments arises endogenously from the behaviour of others in a group. This is important because in order to uncover the design features of social learning strategies, we must study their use in an informational environment they themselves create.

In each session, two groups of four individuals engage in a *four-armed bandit* task [25,26]. Aiming to maximize monetary rewards, participants learn to identify the currently optimal option based on their own returns and the choices of group members. At certain times, participants switch group membership and migrate into the other region having to adopt locally adaptive behaviour. In addition to these spatial changes, the environment also switches temporally altering optimal behaviour in both regions. Using multi-level computational learning models, we are able to infer learning strategies at individual-level resolution. We investigate how individuals strategically use social information and how their social information use differentially responds to migration events (i.e. spatial changes) and temporal changes in the environment. We augment these multi-level models by estimating time-varying parameters through monotonic effects and Gaussian processes and describe how learning dynamics unfold over time. Finally, we employ agent-based simulation to validate statistical models ahead of time and check model predictions by simulating new 'participants' from parameter estimates.

# 2. Methods

## 2.1. Participants

A total of 200 individuals (127 self-identified as women, 73 as men, mean age (s.d.) 30.2 (11.8) years) participated in the social learning experiment (37 additional participants in individual learning control, see below). One hundred and ninety participants named German as/among their first language(s) and everyone was proficient enough to understand instructions. We recruited participants through an institute database, leaflets handed out at Leipzig University and several online advertisements. Written informed consent was obtained prior to the start of the experiment in accordance with the Declaration of Helsinki. Participants received between €12 and €14 in cash straight after the experiment, based upon their performance. No deception was involved in this study and participants were instructed to collect as many points as possible to increase their monetary reward.

## 2.2. Set-up

For each of in total 25 sessions, eight participants were sorted into two random, anonymous groups of four individuals. Each session was conducted by the same experimenter (DD) at the same time of the day in a computer laboratory at the Max Planck Institute for Evolutionary Anthropology in Leipzig, Germany, and lasted between 30 and 60 min depending on the speed of the participants. While participants in the same session were sitting in front of individual computers (Dell OptiPlex 7460 All-in-One desktop computers with 23.8-inch displays) in the same room, they did not know which of the other participants they were sorted into a group with and who the pieces of social information were coming from. Separation walls and enough space between participants ensured they could not see each others' computer displays. The experiment was programmed in oTree v. 2.2.4, a Python-based open-source platform for behavioural research [27]. To transfer information between participants' computers, we set up a local Microsoft Windows 10 web server and used a PostgreSQL database to store the data [28].

## 2.3. Design

In each session, two groups of four individuals engaged in 100 rounds of a four-armed bandit task [25,26]. The experimental task was framed as a farming game (e.g. [8,9]) and participants had to decide in each round to plant one of four different crops (wheat, potatoes, corn, rice, figure 1). Participants were told that both groups live on different sides of a river and, due to differences in the local ecology, different options might be optimal in both regions (in figure 1, corn is currently optimal in region 1 and potatoes in region 2). Which region individuals were currently in was indicated by the background colour of the screen (green versus blue) and was explicitly stated at the top of the display (region 1 versus region 2). Every five rounds, one individual from the first group switched group with a fixed individual from the second group and migrated into the other region. This migration dynamic created a situation in which each individual in a group was characterized by a distinct level of experience in the current region. Groups have completely switched regions after 20 rounds and

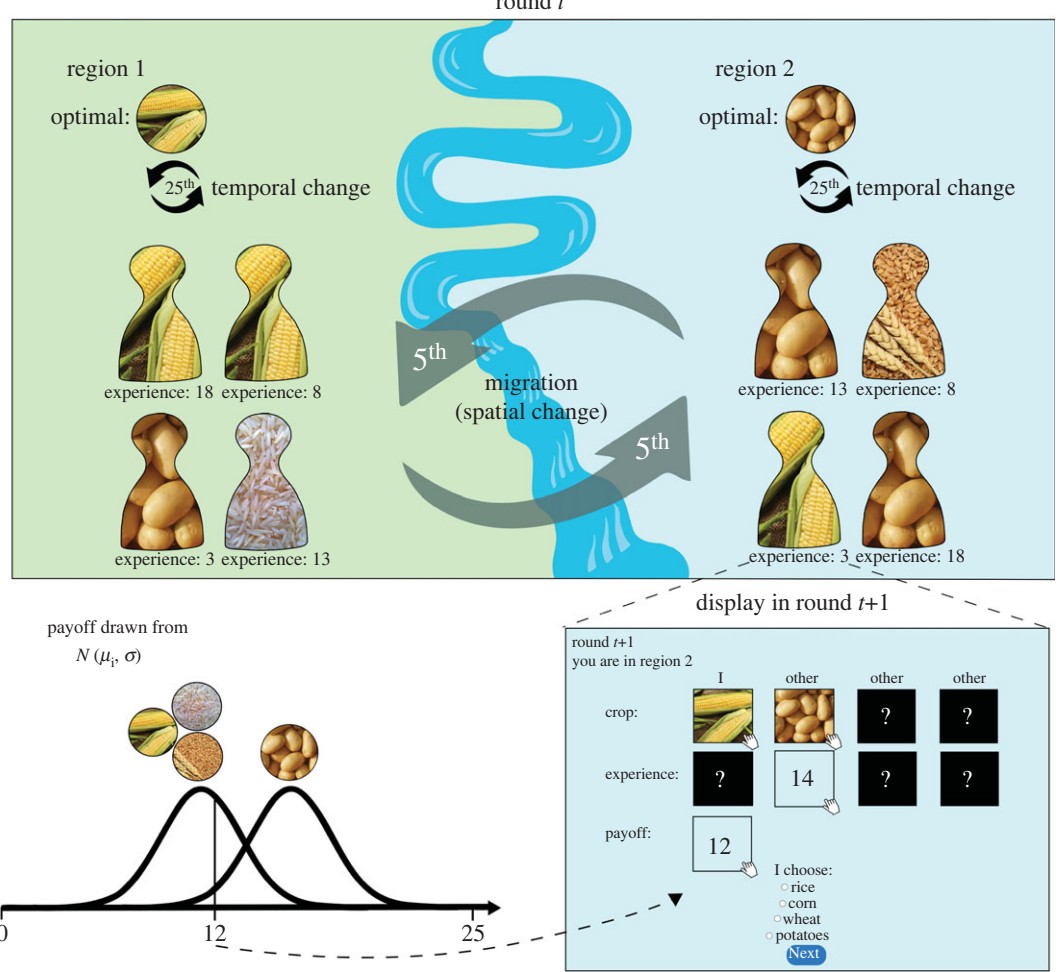

**Figure 1.** Illustration of experimental design and computer display. See Methods section for detailed description.

returned back to their original state after 40 rounds (see electronic supplementary material, figure S1 for full migration schedule).

The experiment was divided into four phases of 25 rounds each and participants were informed that temporal changes, which would affect optimal crops in both regions, might occur between the phases. Which option yielded the highest average payoff in each phase was randomly determined for both groups but in a way to ensure that different crops are optimal in both regions (see in electronic supplementary material, figure S1). Therefore, migrating individual always had to relearn which crop yields the highest payoff in the present environment. Similarly, after a temporal change participants always needed to update their behaviour in order to maximize payoffs. Every time participants migrated and/or reached the end of a phase, they were informed about the total number of points they had collected up to this point and were reminded that a different option might (or might not) become optimal now.

## 2.4. Decision environment

In each round, participants decided between four different options (crops). Payoffs for all options $i$ were randomly drawn from a normal distribution $\mathcal{N}(\mu_i, \sigma)$, so that payoffs varied among rounds but each option was characterized by a given expected value $\mu_i$ (see bottom left part of figure 1). At each point in time and for a given region, one option always had a higher expected payoff than the other three options and participants' task was to find the highest-paying option in order to maximize their payoff. The height of payoffs varied among the four phases of the experiment but the difference between the means of the optimal and the other options was always 3 points (means of 13, 15, 17 and 19 points for the optimal options, respectively). We compared a high task uncertainty condition, in which the payoff

distributions were greatly overlapping ($\sigma = 3$; 'hard' phases), with a low task uncertainty condition, in which there was little overlap ($\sigma = 1.5$; 'easy' phases). Two randomly selected phases were relatively 'hard', the other two were relatively 'easy'.

After the first round, participants could access information about their individual payoff from the previous round (i.e. private information) as well as the crop choices of the other group members from the previous round (i.e. social information) as shown in the bottom right part of figure 1. Additionally, participants could obtain information about how many rounds a given individual (including themselves) has already spent in the current region (experience). We did not include payoff information from other participants, as it is unrealistic to assume that learners can reliably access other individuals' payoffs in most real-world situations. Our experiment thus models only scenarios where this information is either absent or too unreliable. The order in which group members are displayed and the order of crop options to choose from was randomized in each round. To prevent indirect learning about the other region, participants could not see which option newly arrived members chose in the previous round.

## 2.5. Mouse tracking

All information was hidden at first and individuals must hover over the boxes to see the respective information (see bottom right part in figure 1). We used *mouselab* [29] JavaScript to record all occasions when individuals entered and left a box and calculated from that raw data the time per round that an individual spent in each box. That way, we can not only study how participants sample different sources of information but also have complete data about the informational environment that resulted in a given behavioural choice. Below, we use detailed mouse-tracking information to condition learning strategies on the sources of social information individuals actually accessed in each round. Analysing full search strategies using for instance information foraging models [30] would be a fruitful avenue for future research.

## 2.6. Individual learning control

To compare learning in the social learning experiment with an asocial control and to ensure there was no difference between spatial and temporal changes arising from framing effects or other confounds, we let 37 additional individuals (18 self-identified as women, 19 as men, mean age (s.d.) 29.9 (13.8) years) perform an individual learning condition. The procedure was identical to the one described above with the only exception that participants were not assigned to a group and thus could only see their own choice and payoff from the previous round.

## 2.7. Data analysis

### 2.7.1. Experience-weighted attraction models

Looking to infer learning strategies from behaviour, we are faced with a so-called inverse problem, i.e. going from (overt) observations to (hidden) causes. This constitutes a problem because typically many different processes can result in the same empirical pattern [31,32]. By formulating scientific models as statistical models, however, we can estimate which parameter values are most compatible with the observed choices [33]. Here, we use Bayesian multi-level experience-weighted attraction (EWA) models that link individual (reinforcement-learning) updating rules and social information to population-level cultural dynamics [8,34,35].

There are two basic components: first, we have an updating or learning equation that tells us how *attractions* to different behavioural options $A_{i,j,t+1}$ (i.e. how preferable option $i$ is to the actor $j$ at time $t+1$) change over time as a function of previous attractions $A_{i,j,t}$ and recently experienced payoffs $\pi_{i,j,t}$. The (participant-specific) parameter $\phi_j$ describes the weight of recent experience. The higher the value of $\phi_j$, the faster do learners update their attractions

$$A_{i,j,t+1} = (1 - \phi_j)A_{i,j,t} + \phi_j \pi_{i,j,t}. \tag{2.1}$$

The second major part expresses the probability an individual $j$ chooses option $i$ in the next round, $t+1$, based on a series of cues. We can divide those cues into asocial ($P_A$) and social cues ($P_S$) and the model

lets us estimate the relative influence of social versus asocial cues ($\sigma_j$):

$$P(i|A_{i,j,t}, \theta_j)_{t+1} = (1 - \sigma_j)P_{A,i,j,t+1} + \sigma_j P_{S,i,j,t+1}. \tag{2.2}$$

The asocial choice probability $P_A$ is determined by a multinomial-logistic or *softmax* choice rule which translates the attraction towards option $i$ into the probability this option is chosen in the next round:

$$P_{A,i,j,t+1} = \frac{\exp(\lambda_j A_{i,j,t})}{\sum_{m=1}^{4} \exp(\lambda_j A_{m,j,t})}. \tag{2.3}$$

The parameter $\lambda_j$ represents the exploration rate of an individual (also called *inverse temperature*). It controls how sensitive choices are to differences in attraction scores. As $\lambda_j$ gets larger, choices become more deterministic, as it gets smaller, choices become more exploratory (random choice if $\lambda_j = 0$).

Individuals in the experiment have access to different sorts of social information. Our model estimates the relative influence of conformist ($P_C$) and experience-directed ($P_E$) learning as a convex combination of both cues with parameter $\kappa_j$ giving the relative influence of experience cues relative to frequency cues

$$P_{S,i,j,t+1} = (1 - \kappa_j)P_{C,i,j,t+1} + \kappa_j P_{E,i,j,t+1}. \tag{2.4}$$

The frequency-dependent or conformist probability is given by

$$P_{C,i,j,t+1} = \frac{n_{i,t}^{f_j}}{\sum_{m=1}^{4} n_{m,t}^{f_j}}, \tag{2.5}$$

where $n_{i,t}$ represents the number of group members that chose option $i$ in the previous round. Conformity exponent $f_j$ determines how strongly learning is biased towards the majority. When $f_j = 1$, learning is unbiased; as $f_j$ becomes larger, individuals become more and more likely to copy the majority. When $0 < f_j < 1$, individuals are disproportionately copying the minority option. The experience-biased probability is given by

$$P_{E,i,j,t+1} = \frac{\sum_{k=1}^{n_{i,t}} \exp(\beta_j E_{k,t})}{\sum_{m=1}^{4} \sum_{k=1}^{n_{m,t}} \exp(\beta_j E_{k,t})}, \tag{2.6}$$

where $E_{k,t}$ gives the experience of the $k$th of $n_{i,t}$ group members who chose option $i$. Parameter $\beta_j$ determines the strength and direction of experience bias. When $\beta_j = 0$, individuals are indiscriminate with respect to experience in the current region. Negative values of $\beta_j$ indicate a bias towards less-experienced individuals, positive values a bias towards more-experienced individuals. This parametrization provides a straightforward way to combine conformist learning that operates on the basis of choices and experienced-biased learning that operates on the level of individuals. We validated this approach by simulating data from different parametrizations and ensuring this model would recover simulated dynamics (see preregistration or electronic supplementary material for details).

### 2.7.2. Time-varying learning parameters

To investigate how learning unfolds over time after migration, we included temporally dynamic learning parameters. Note we only describe $\sigma$, the weight of social learning, in detail here, but other learning parameters were constructed in the same way. We took two approaches. First, instead of imposing a particular function, we only assumed that learning parameters change monotonically over time, i.e. either constantly decrease or increase, and let the model estimate the size of the steps in which learning strategies change. The value for $\sigma_j$ after $\ell$ rounds in a new region can be expressed as follows:

$$\sigma_{j,\ell} = \sigma_{t_{\min},j} - (\sigma_{t_{\min},j} - \sigma_{t_{\max},j}) \sum_{m=0}^{\ell-1} \delta_m. \tag{2.7}$$

Each individual is characterized by two parameters, $\sigma_{t_{\min},j}$ and $\sigma_{t_{\max},j}$ representing values for the shortest and longest time since migration, respectively. These values determine how much $\sigma$ changes over time for individual $j$, their difference thus represents the total effect of time. This total effect is multiplied by the sum of a number of $\delta$ parameters which give the incremental effect of each additional time step (note that $\delta_0 = 0$ and all $\delta$ parameters must sum to one). To allow for sudden shifts in learning, which are realized through large differences in $\delta$ values, we chose a relatively weak Dirichlet prior (with $\alpha = 2$) for the vector of $\delta$ parameters [33].

Second, we modelled the effect of time since migration as a Gaussian process, where the model can estimate any arbitrary function. Gaussian processes extend the varying effects approach to continuous categories and estimate a unique parameter value for each level, while still regarding time as a continuous dimension in which similar levels result in more similar behaviour [33]. Specifically, the value of $\sigma$ after $\ell$ rounds in a new region is composed of the average across rounds and a round-specific offset: $\sigma_\ell = \bar{\sigma} + d_\ell$. We define a multivariate Gaussian prior (hence the name 'Gaussian process') for the round-specific offsets $d_\ell$

$$\begin{pmatrix} d_1 \\ d_2 \\ \dots \\ d_{20} \end{pmatrix} \sim N \left[ \begin{pmatrix} 0 \\ 0 \\ \dots \\ 0 \end{pmatrix}, K \right]. \tag{2.8}$$

The vector of means is all zeros, so the average weight of social learning remains unchanged, and $K$ is the $20 \times 20$ covariance matrix among levels of experience. We estimate the parameters of a function that expresses how the covariance between different levels is expected to change as the distance increases

$$K_{x,y} = \eta^2 \exp(-\rho^2 D_{x,y}^2) + \delta_{x,y} \sigma^2. \tag{2.9}$$

The covariance between any pair of times $x$ and $y$, $K_{x,y}$, equals the maximum covariance $\eta^2$ which is reduced at rate $\rho^2$ by the squared distance in time between $x$ and $y$, $D_{x,y}^2$. There is an additional covariance parameter $\sigma^2$ that gets 'turned on' by $\delta_{x,y}$ when $x = y$; it expresses the additional covariance for observations with the same time since migration. With only a handful of replicates per individual for each time interval, we could not fit the fully multi-level Gaussian process model that estimates participant-specific variance-covariance matrices, so we report trends across all individuals below.

All models were fitted using the Hamiltonian Monte Carlo engine Stan [36] in R v. 3.6.0 [37] from Rstan v. 2.19.2 [38]. We used weakly informative normal priors centred on 0 for learning parameters, which were estimated on the linear ($\beta$), logarithmic ($\lambda$, $f$) and logit ($\sigma$, $\kappa$, $\phi$) scale, respectively, exponential priors for scale parameters and Lewandowski–Kurowicka–Joe (LKJ) priors for correlations matrices [39]. To improve convergence, we implemented the non-centred version of varying effects using a Cholesky decomposition of the correlation matrix [33]. For all analyses, visual inspection of traceplots and rank histograms [40] suggested good model convergence and no problematic autocorrelation, with convergence confirmed by the Gelman–Rubin criterion $\hat{R} \leq 1.01$ [41]. All inferences are based on over 1000 effective samples from the posterior [42]. The code necessary to reproduce all analyses is available on GitHub (https://github.com/DominikDeffner/Dynamic-Social-Learning).

### 2.7.3. Pre- and post-experimental simulations

To validate our analytical approach before data collection, we conducted agent-based simulations using the exact same set-up and parameter values we used for the actual experiment. In 25 sessions, we followed eight simulated 'participants' through 100 rounds of the experiment and recorded their choices in a comparable format. The behaviour of agents was governed by the same mathematical learning rules we used for the statistical models. These simulations allowed us to (i) choose good experimental design parameters (round number, migration rate, difficulty etc.), and (ii) verify that the models recover simulated parameter values in both extreme and more realistic scenarios. Electronic supplementary material, figure S2 shows exemplary results of the parameter recovery test for the monotonic-effects model. Half of simulated agents relied heavily on social learning in the beginning and then switched completely to individual learning, while the rest relies on intermediate amounts of social learning irrespective of experience. We also implemented sudden shifts in agents' conformity and experience bias. The model accurately recovered strategies in both sub-groups and also produced quantitatively matching parameter estimates. Further information on our simulations are described in the preregistration at https://osf.io/j6v5m/ or in the electronic supplementary material.

Simulations are critical to plan experiments and validate statistical models ahead of time, but they can also be used after data collection to generate novel predictions from parameter estimates. Multi-level models adaptively regularize individual parameters and the covariances among them by estimating a population of varying effects typically defined as a multivariate normal distribution. Maintaining the correlation structure among parameters, we can draw samples from that distribution and simulate new 'participants'. As the model is trained to predict choices in the next round based on detailed time-series data and not to fit whole learning trajectories, we cannot expect our simulated participants to behave in the same way real human participants did. Inspecting how the behaviour of simulated

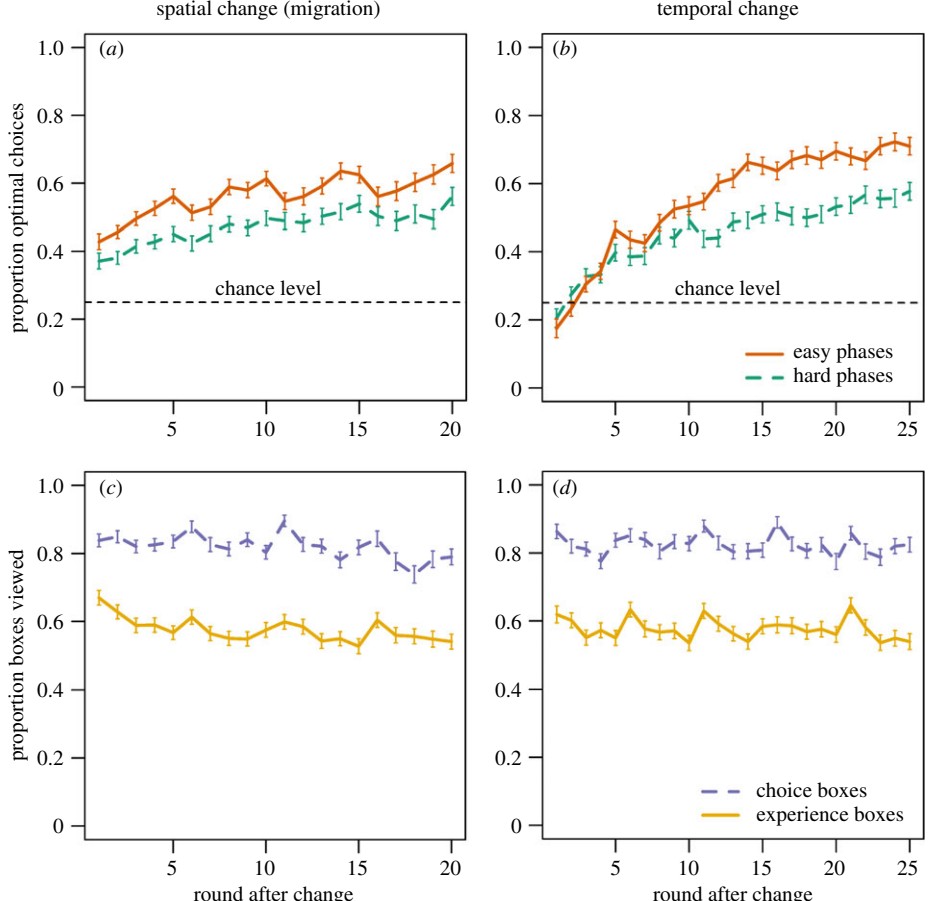

**Figure 2.** Behavioural results: proportion of optimal choices per round after (a) spatial change (migration) and (b) temporal environmental change. Solid orange lines represent relatively easy phases ($\sigma = 1.5$), dashed green lines represent relatively hard phases ($\sigma = 3$). Dashed horizontal lines show chance level. Proportion of social information boxes (choice boxes dashed purple, experience boxes solid yellow) viewed per round after (c) spatial change (migration) and (d) temporal environmental change. Error bars show standard errors of the means (s.e.m., $\sigma_{\bar{x}} = s/\sqrt{n}$).

participants diverges from the real participants, however, provides valuable insights into the implications of the model that are invisible from the parameter estimates alone. Full simulation code can be found on GitHub (https://github.com/DominikDeffner/Dynamic-Social-Learning).

# 3. Results

## 3.1. Behavioural results

### 3.1.1. Learning

Participants learned to choose the optimal option over time after migration into a new region (figure 2a) and after a temporal change in the environment (figure 2b). In the first round after migration, individuals already performed considerably better than chance (25%) which was probably supported by social learning from residents possessing optimal behaviour (see below). Spending more time in a region generally increased proportions of optimal choices confirming our expectation that more experienced individuals should overall show better performance. However, there are drops in adaptive behaviour after 5, 10 and 15 rounds, respectively, that are due to occasional temporal changes occurring at these time intervals after migration. In contrast to spatial changes, temporal changes initially resulted in choices below chance indicating that individuals first continued with their previous choices. The drops in adaptive behaviour due to migration events are less pronounced, as social information from

**Table 1.** Results of baseline multi-level EWA model. Posterior means, 89% highest posterior density intervals (HPDI), standard deviations of varying effects across individuals and correlations with overall success across individuals.

| parameter | interpretation | post. mean | 89% HPDI | s.d. individuals (HPDI) | corr w/ payoffs (HPDI) |
|---|---|---|---|---|---|
| $\sigma$ | weight of social learning $(0 \rightarrow 1)$ | 0.29 | 0.27–0.32 | 0.76 (0.66–0.86) | 0.35 (0.29–0.41) |
| $\kappa$ | weight of experience bias $(0 \rightarrow 1)$ | 0.21 | 0.07–0.36 | 1.67 (1.02–2.43) | 0.05 (−0.09–0.19) |
| $f$ | conformity exponent $(0 \rightarrow \infty)$ | 3.30 | 2.21–4.35 | 0.50 (0.07–1.02) | 0.01 (−0.19 − 0.20) |
| $\beta$ | experience bias $(-\infty \rightarrow \infty)$ | 0.50 | 0.10–0.96 | 0.33 (0.11–0.72) | 0.07 (−0.13 − 0.26) |
| $\lambda$ | (inverse) exploration rate $(0 \rightarrow \infty)$ | 0.13 | 0.12–0.14 | 0.30 (0.26–0.35) | 0.27 (0.16–0.37) |
| $\phi$ | updating/learning rate $(0 \rightarrow 1)$ | 0.72 | 0.64–0.79 | 2.42 (2.02–2.87) | 0.18 (0.13–0.24) |

residents buffered the effect of spatial changes. Participants in an individual learning control condition (electronic supplementary material, figure S3) learned more slowly and exhibited no clear difference between spatial and temporal changes, suggesting that the observed patterns in the social learning experiment are indeed due to different social environments.

### 3.1.2. Social information boxes

The bottom row in figure 2 shows the proportion of social information boxes individuals inspected conditional on round after migration (figure 2c) and temporal change (figure 2d). Overall, individuals viewed 75–90% of boxes showing previous choices of other group members and 50–65% of boxes showing their respective levels of experience in the present region. While rates of social information use mildly declined over time, participants tended to search for more social information at times when temporal changes occurred or they migrated into the other region suggesting a 'copy-when-uncertain' strategy [1,5,6].

## 3.2. Computational modelling

### 3.2.1. Baseline model

As a first step, we constructed a multi-level EWA model where learning parameters varied by individual but did not change across rounds of the experiment. Table 1 shows posterior means, 89% highest posterior density intervals (HPDI), standard deviations of varying effects and correlations with overall success across individuals for all major model parameters [33]. Individuals used social information to guide their choices ($\bar{\sigma} = 0.29$; HPDI $= 0.27 - 0.32$) with considerable variation among individuals. We have seen in the previous section that, due to increased learning opportunities, more experienced individuals tended to show higher proportions of optimal choices. In line with this distribution of adaptive behaviour, participants preferentially copied more experienced rather then less experienced individuals ($\bar{\beta} = 0.50$; HPDI $= 0.10 - 0.96$) exhibiting what could be called an 'experience-biased' social learning strategy [16]. As reported in previous experiments [8,9,11], participants disproportionately copied the most common option among observed neighbours exhibiting a pronounced 'conformity' bias ($\bar{f} = 3.30$; HPDI $= 2.21 - 4.35$). Participants promptly updated their behaviour in light of new experiences ($\bar{\phi} = 0.72$; HPDI $= 0.64 - 0.79$) which reflects the various environmental changes individuals were confronted with. From the posterior distributions of each participant-specific parameter, we can calculate how learning strategies were associated with overall payoffs. This analysis reveals that participants who relied more heavily on social information ($r_{\sigma,\text{Pay}} = 0.35$; HPDI $= 0.29 - 0.41$),

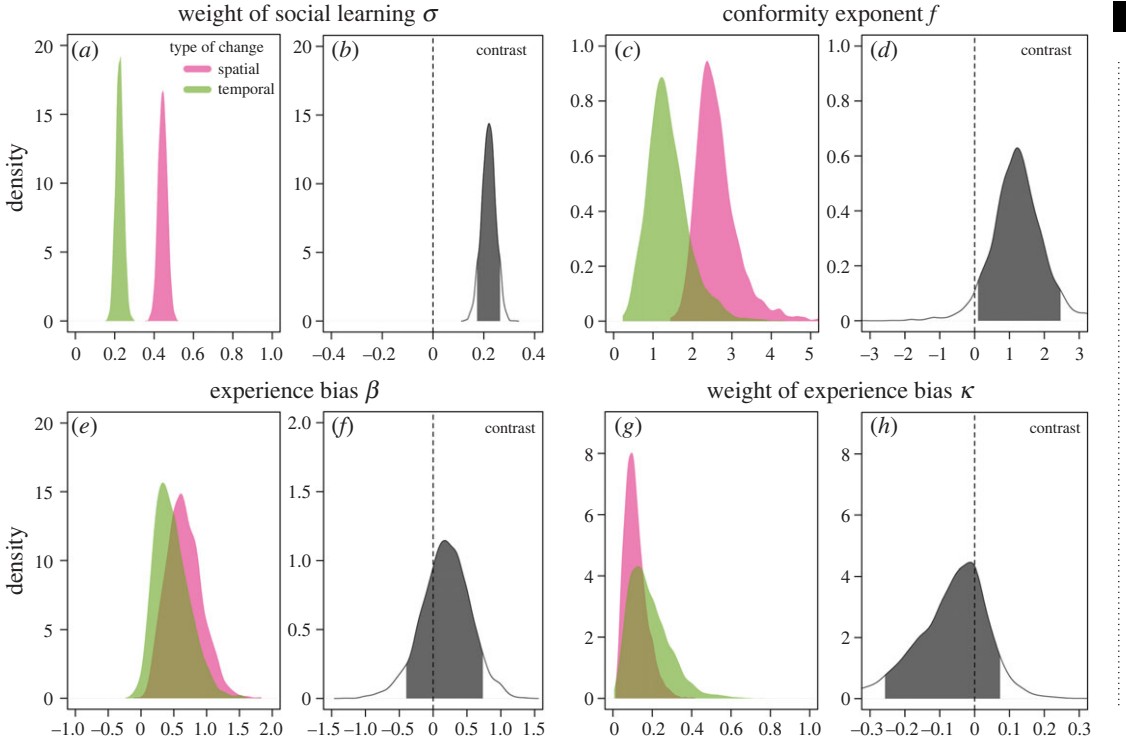

**Figure 3.** Comparison of learning for first five rounds after spatial (pink) and temporal (green) changes in the environment: marginal posterior probability distributions for (a) $\sigma$, the relative weight of social versus individual learning, (c) $f$, the strength and direction of frequency bias (conformity), (e) $\beta$, the strength and direction of experience bias and (g) $\kappa$, the relative weight of experience bias versus frequency bias. Plots (b), (d), (f) and (h) show posterior distributions (including 89% HPDI) for respective contrasts between spatial and temporal changes.

who updated their behaviour more quickly in response to recent payoffs ($r_{\phi,\text{Pay}} = 0.18$; HPDI $= 0.13 - 0.24$) and who let their attraction values more strongly determine their choices ($r_{\lambda,\text{Pay}} = 0.27$; HPDI $= 0.16 - 0.37$) tended to collect more points in the experiment (using the geometric mean corresponding to multiplicative fitness effects gives the same results). Social information use was relatively low compared with individual learning with enough group members tracking the state of the environment. Therefore, social learners could benefit from this accumulated collective knowledge and improve their performance relative to more individual learners [4,43].

### 3.2.2. Temporal versus spatial changes

To directly compare learning strategies after temporal and spatial changes, we repeated the previous analysis but included indicator or dummy variables that let us compute contrasts in learning parameters between the first five rounds after spatial changes and the five first rounds after temporal changes. Results revealed that participants relied substantially more on social learning after spatial changes compared with temporal changes (figure 3a,b). Migrants enter established groups where other members already had multiple rounds to learn the optimal option increasing the value of social learning. Temporal changes, by contrast, result in a situation where all group members become non-adapted and need to learn the new optimal solution (only 31% of group members chose the optimal option in the first five rounds after a temporal change compared with 53% after a spatial change). Under these circumstances, it is beneficial to rely more on individual learning. As expected from the modelling results in [18], participants were more likely to copy the majority option after spatial changes compared with temporal changes (figure 3c,d). While spatial changes resulted in a clear signal of conformist social learning with all posterior probability lying above 1 (which represents unbiased copying), learning after temporal changes was not clearly conformist with substantial posterior probability lying around and below 1. Experience cues should only be correlated with rates of adaptive behaviour after spatial changes, but not after temporal changes. Therefore, we expected participants to rely more heavily on experience cues after spatial compared with temporal changes.

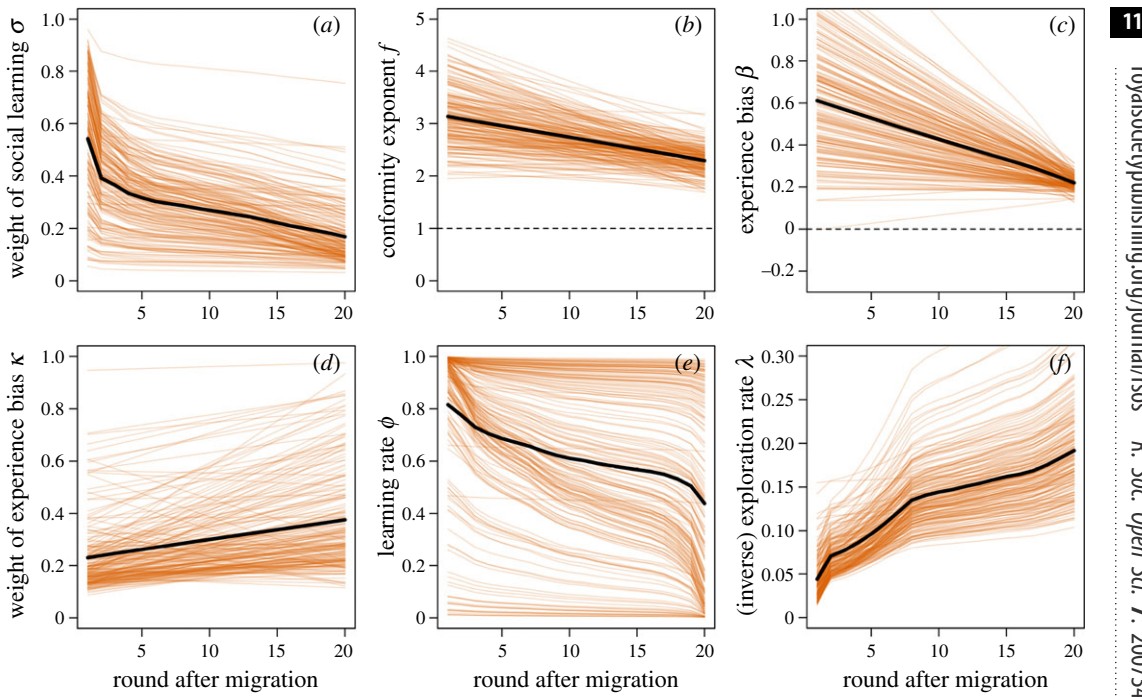

**Figure 4.** Monotonic effects: dynamic learning strategies after migration. (a) $\sigma$, the relative weight of social versus individual learning, (b) $f$, the strength and direction of frequency bias (conformity), (c) $\beta$, the strength and direction of experience bias, (d) $\kappa$, the relative weight of experience bias versus frequency bias, (e) $\phi$, the learning/updating rate and (f) $\lambda$, the (inverse) exploration rate. Orange lines show posterior estimates for each of 200 participants and black lines show means over all participants.

Although most of the posterior density of the contrast points into this direction, there was no distinct difference in the reliance on experience information with individuals being more likely to copy more experienced individuals irrespective of the type of environmental change (figure 3e,f). Similarly, there was also no distinct difference in the relative weight placed on experience versus frequency information (figure 3g,h). In sum, these results suggest that participants adaptively adjusted social learning strategies broadly in line with predictions from theoretical models.

### 3.2.3. Time dynamics of strategic social learning

Next, we included time-varying learning parameters on top of individual-specific varying effects to explore how learning dynamically changed over time. Figure 4 shows how learning parameters changed after migration into a new region according to the multi-level monotonic-effects model. Right after migration, most individuals relied very heavily on social information from residents in the new region ($\bar{\sigma} \approx 0.55$; figure 4a). The mean social learning weight then halves after approximately five rounds and falls below 0.2 eventually. The results also reveal large inter-individual variability with some participants almost exclusively relying on individual information throughout the experiment. As participants spent more time in a region, they seem to become slightly less conformist over time (figure 4b) and their tendency to copy more experienced group members also declined (figure 4c). Because individuals relied on relatively small amounts of social learning and inspected only around 50% of experience boxes (figure 2c) after 20 rounds in a region, the model struggled to estimate the variation among participants for $\beta$ long after migration resulting in the strong convergence of estimates. Over time, experience information became marginally more important relative to frequency information (figure 4d). Finally, the rate at which individuals updated their beliefs declined over time (figure 4e) and individuals became more sensitive to differences in attraction scores, i.e. less exploratory (figure 4f).

Reassuringly, without making any assumptions about the shape of the functions, the Gaussian processes largely confirm the previous results but also add some interesting nuance (figure 5). Social information use is highest in the beginning, drops significantly in the first few rounds after migration and then declines at a relatively constant rate. Conformity is also highest straight after migration, then drops but remains relatively high throughout the experiment. As parameters in EWA and other learning models can interact in highly nonlinear ways, we also report results for analyses where we

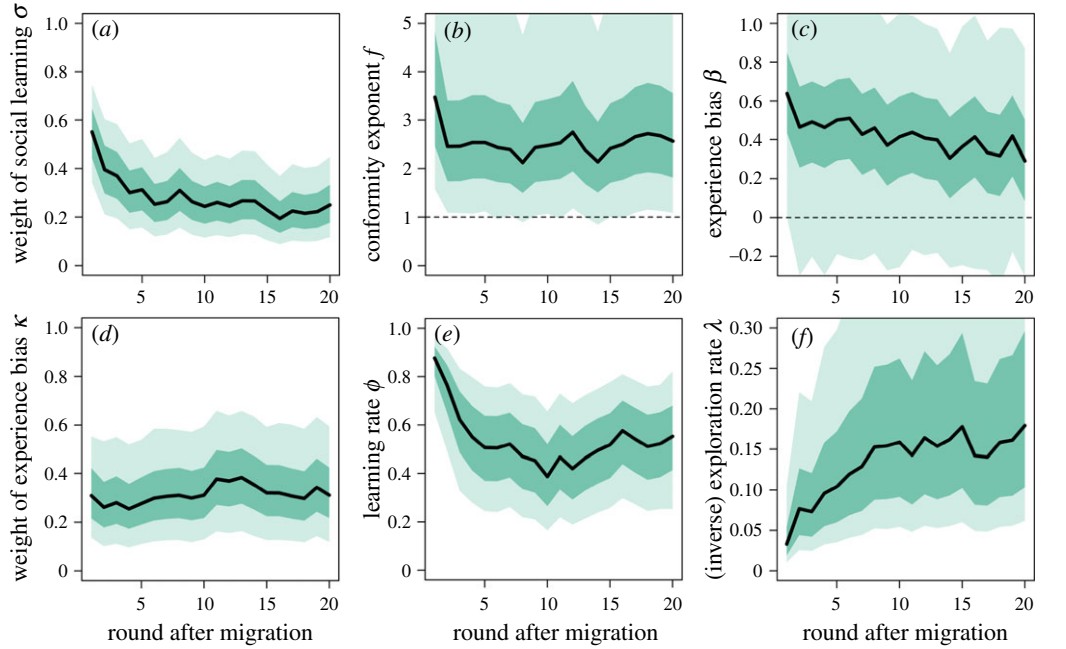

**Figure 5.** Gaussian processes: dynamic learning strategies after migration. (*a*) $\sigma$, the relative weight of social versus individual learning, (*b*) *f*, the strength and direction of frequency bias (conformity), (*c*) $\beta$, the strength and direction of experience bias, (*d*) $\kappa$, the relative weight of experience bias versus frequency bias, (*e*) $\phi$, the learning/updating rate and (*f*) $\lambda$, the (inverse) exploration rate. Dark shaded areas show 89% HPDIs around the mean, light shaded areas include 89% HPDIs around round-specific deviations.

let only social learning parameters change over time while holding others constant (see electronic supplementary material, figures S4 and S5 for monotonic effects and Gaussian processes, respectively). These results largely confirm results from fully time-varying models reported here.

### 3.2.4. *Post hoc* simulations

To better understand the implications of the computational learning models, we conducted agent-based simulations of the experiment with 200 new 'participants' sampled from the estimated population of varying effects. Electronic supplementary material, figure S6 shows learning curves after spatial and temporal changes analogous to figure 2 that plots data from human participants. We compare the behaviour of agents simulated from results of the baseline model where learning parameters are constant over time (top row) to agents simulated from the time-varying monotonic-effects model (bottom row). Overall, simulated 'participants' learned much slower than their human counterparts. Only by increasing the difference between expected payoffs we obtain remarkably similar patterns to real participant, especially for agents simulated from the time-varying model (see electronic supplementary material, figure S7). This highlights that our model is most definitely missing some important cognitive detail typical of real human participants. Unlike simulated agents, participants were told that one option is optimal at each point in time, so that points were presumably used to update beliefs about which option is best in a categorical way instead of the continuous nature of pure reinforcement learning.

## 4. Discussion

We investigated strategic social learning in dynamic groups including both spatial and temporal variability. To understand how culture evolves and operates in real organisms, it is not enough to study the dynamics of learning and cultural information in isolation. Instead, we need more theoretical and empirical work that investigates how learning intersects with demography and population dynamics and flexibly responds to different informational environments [14]. As a step in this direction, we designed a laboratory experiment where two groups of four individuals (per session) learned locally optimal behaviour in a four-armed bandit task. Participants occasionally

migrated between two regions (spatial changes) and also were faced with temporal changes in the environment. We used computational learning models to identify individual-level strategies and included time-varying parameters to explore how participants strategically adjusted learning over time. Experiments are gross simplifications of reality. They abstract away from most real-world complexities and result in a highly idealized version of the phenomena they aim to represent. Good experiments, however, also lay bare the fundamental structure of an otherwise overly complex system and confront participants with controlled situations that elicit behavioural strategies that are hard or even impossible to observe in more naturalistic settings. Well-designed experiments thus resemble theoretical models in that their simplification is a critical design feature and not a bug.

We found that, overall, individuals used both experience and frequency cues to direct their social information use. In the social learning literature, a 'copy older over younger models' strategy has been suggested to be adaptive because older individuals have had more experience with the environment. Empirical tests that mostly let children decide to either copy an adult or a child demonstrator generally confirmed this prediction [16]. Such studies, however, cannot isolate experience as the relevant factor nor are there any adaptive consequences to the choices participants make. Our results demonstrate that experience can endogenously arise as a predictive cue of successful behaviour that human participants actually use to selectively learn behaviour. Being particularly old or experienced may often be a signal of having adaptive behaviour but it can also predict being out of date, especially if there is an exploration–exploitation trade-off and older individuals are less likely to update their behaviour. Future experiments could test this idea by implementing a learning cost or forcing participants to either learn or exploit a known option in every given trial.

We also found that participants selectively responded to different changes in the environment. After spatial changes (i.e. migration events), individuals heavily relied on conformist social learning which let them quickly and reliably adopt the locally adaptive behaviour. After temporal changes that render everyone non-adapted at the same time, by contrast, participants relied more on individual learning and were less conformist in their social information use. These findings support important but hitherto untested predictions of the theoretical cultural evolution literature [4,18,44]. Rates and strategies of social information use in humans and other animals can also be expected to vary outside the laboratory depending on the dominant mode of environmental variation. This prediction could be tested by comparing social learning in similar communities that are characterized by more or less spatial structure and/or different rates of temporal environmental change.

Social learning tendencies in most animals probably are not fixed traits, but adjustable propensities that can flexibly respond to changing ecological conditions. To account for both stable inter-individual differences in social learning as well as changes depending on situational factors, we estimated time-varying learning parameters on top of participant-specific varying effects. These analyses revealed that participants relied on very high levels of social learning straight after migration with rates of social information use dropping as individuals became more experienced in their new region. Conformist tendencies remained relatively stable over the course of the experiment. These results potentially help explain which factors could maintain between-group cultural diversity in the face of frequent migration that would otherwise erode cultural differences. It has been suggested that conformist social learning can stabilize between-group cultural variation because it makes migrants quickly acculturate to community-typical norms [22]. Our findings corroborate this expectation and add that relatively stable conformist tendencies of resident individuals similarly act to filter out cultural variation brought in by newly arrived group members thereby also reducing within-group cultural diversity. Understanding the mechanisms underlying such structured cultural variation is not only important to understand culture itself but also has important implications for cultural group selection, a proposed explanation for human large-scale cooperation [45,46]. This theory presupposes that alternative equilibria of local norms and behaviours are stabilized through learning processes in different groups, and group-level selection between such culturally differentiated populations might then lead to the spread of cooperative social norms that benefit the cultural group.

We used computational learning models to investigate how strategies varied among individuals and changed over time. Through *post hoc* simulation, we found that simulated agents showed similar overall patterns of behaviour but learned much slower than real human participants. Implementing more realistic individual learning models from computational neuroscience based on, for example, variational inference could be fruitful avenues for future research [47–49]. Additionally, the way our model estimates time-varying learning parameters through monotonic effects and Gaussian processes is purely stochastic, i.e. relying on statistical associations, and not mechanistic, i.e. grounded in cognitive processing. Alternatively, in hierarchical Gaussian filter models, which have also been

applied to social contexts, learning parameters can vary depending on the perceived reward variability of the environment potentially linking observed changes in learning strategies to underlying cognitive mechanisms [50,51].

Studies in the behavioural and social sciences have been repeatedly criticized for relying almost exclusively on WEIRD samples [52]. Although not the typical student sample, participants in our study were also predominantly Western, relatively educated, and from an industrialized, rich, and democratic country. While we do not find the WEIRD/non-WEIRD dichotomy particularly useful, as it hides important cultural variation both within and between societies, generalizability of empirical findings to new environments, settings or populations is always a concern for experimental studies. This concern is typically addressed by (calls for) replication across different conditions and populations. Although replication across societies may suggest greater generalizability, it is impossible to sample all relevant populations, and drawing inferences from either positive or negative results is difficult without a profound theoretical understanding of factors expected to cause cross-cultural differences and their distribution across populations. As a complement to this bottom-up, data-driven approach, researchers in computer science have put forward a rigorous formal framework for licensed transfers of causal effects from experimental studies to new populations, in which only observational studies can be conducted [53,54]. Developing a similar framework for cross-cultural generalizability would be a major advance.

Human evolution has been characterized by massive range expansions and constant migration events between different populations [55,56]. To understand how culture evolved in our ancestors and how it facilitates flexible human adaptation, we need to develop more formal theory and empirical tests of the mechanisms underlying cultural evolution in such highly dynamic scenarios. We provide experimental insights and introduce modelling tools that hopefully can be applied to understand the adaptive logic of dynamic social learning in different systems.

Ethics. The experimental procedure was approved by the HBEC Department Ethics Board at the Max Planck Institute for Evolutionary Anthropology.

Data accessibility. Data and relevant code for this research work are stored in GitHub: https://github.com/DominikDeffner/Dynamic-Social-Learning, and have been archived within the Zenodo repository: https://doi.org/10.5281/zenodo.4034787.

Authors' contributions. D.D. and R.M. jointly designed the work. D.D. programmed and conducted the experiment. V.K. and D.D. recruited the participants. D.D. analysed the data and wrote the first draft of the manuscript. All authors contributed to the final version.

Competing interests. We declare we have no competing interests.

Funding. The research has been funded by the Max Planck Society.

Acknowledgements. We thank Peter Frölich for invaluable help in setting up the laboratory and web server, recruiting participants for pilot experiments as well as constant technical and emotional support. We thank members of the Department of Human Behaviour, Ecology and Culture for testing software and providing critical input in the design stage of this study. We also thank our pilot and study participants for taking their time and supporting our research. Finally, we thank the Max Planck Society for funding.

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
