## [Reviewer comments · Royal Society Open Science]

Review History

RSOS-200734.R0 (Original submission)

Review form: Reviewer 1

Is the manuscript scientifically sound in its present form?

Yes

Are the interpretations and conclusions justified by the results?

Yes

Is the language acceptable?

Yes

Do you have any ethical concerns with this paper?

No

Have you any concerns about statistical analyses in this paper?

No

Recommendation?

Accept with minor revision (please list in comments)

Comments to the Author(s)

See doc (Appendix A).

Review form: Reviewer 2

Is the manuscript scientifically sound in its present form?

Yes

Are the interpretations and conclusions justified by the results?

Yes

Is the language acceptable?

Yes

Do you have any ethical concerns with this paper?

No

Have you any concerns about statistical analyses in this paper?

No

Recommendation?

Accept as is

Comments to the Author(s)

This paper investigates social learning in temporally and spatially variable environments using lab experiments and computational learning models. The results show that individuals use both experience and frequency cues to direct their social information use. Interestingly, the results also show that participants selectively respond to different changes in the environment with more conformist social learning following spatial changes and more individual learning following temporal changes. This latter result confirms previously untested theoretical predictions about how individuals should strategically use social information.

Overall, I found the paper of outstanding quality and I think the conclusions are an important contribution to the field of cultural evolution. The (preregistered) experiment is extremely well designed, the reporting of the data and methodology are very detailed and transparent to enable reproducing the results. Note that I cannot comment on computational learning models which are beyond the scope of my expertise. However, the strategy seems sound to me and, from the parts that I was able to follow, I find that the conclusions drawn from the results are valid. The paper is also extremely well written, which makes even the technical sections easy to read.

I have a couple of minor comments/questions, but overall, I think the paper can be published as is.

Given that experimental sessions "lasted between 30 and 60 minutes depending on the speed of participants", I assume that participants did not face any time constraints to make their decisions. Under those conditions how to explain that participants only viewed 75-90% of boxes showing previous choices of other group members and 50-65% of boxes showing their respective levels of experience in the present region? In absence of costs associated with social information use, I would expect participants to reveal as much of social information as they can (especially because participants can only reveal up to 6 pieces of social information).

In the methods section, the Mouse Tracking subsection states that participants had to hover over the information boxes to see their "content". I assume information was not visible anymore once

participants moved the mouse elsewhere, is that correct? Could working memory limitations explain why participants did not tend to reveal the content of 100% of boxes?

In figure 2, y-axes range from 0.1 to 0.7 in A-B and 0.3 to 1 in C-D. This is slightly confusing, especially because the proportion of optimal choices in B seems to plateau. Distracted readers might incorrectly assume that it is because individuals reach 100% accuracy.

Review form: Reviewer 3 (Ulf Toelch)

Is the manuscript scientifically sound in its present form?

Yes

Are the interpretations and conclusions justified by the results?

Yes

Is the language acceptable?

Yes

Do you have any ethical concerns with this paper?

No

Have you any concerns about statistical analyses in this paper?

No

Recommendation?

Accept with minor revision (please list in comments)

Comments to the Author(s)

Defnner et al. describe in their manuscript "Dynamic Social Learning in Temporally and Spatially Variable Environments" a 4 armed bandit task where participant behaviour analysed via a computational model. The experiment and modelling is well planned and executed, particularly, the approach to first simulate data, then create the experiment, and later perform parameter recovery is something I would wish for in all computational modelling papers.

The methods are sound, but I have only briefly looked at the underlying code and have not performed an in depth review of it.

I have some comments the authors may want to consider.

1. Code of experimental software

Perhaps I missed it, but is it possible to make the oTree code available? I am not overly familiar with the software, but to allow replication of your experiment this would be helpful.

2. Adaptation of parameters.

The experimental environment includes often a high degree of uncertainty. Learning rates and other parameters are thus adjusted.

The authors adjust for this by a. modelling monotonic effects and also Gaussian processes.

This modelling step is underwritten. From the information in the Methods it was not possible for me to follow what the authors were doing but I had to resort to code and results sections. More detail is needed here. How are the actual parameters from Figure 4 related to the equation 2.7.

One example is the use of sigma in the monotonic models. In a first read through I thought this would refer to the balance between the social and individual information.

I also thought that this would be the only parameter that would be adjusted. Please add more information on the modelling process directly in the main manuscript.

3. Model limitations.

The adaptation of parameters is stochastic and not mechanistic/cognitive. That is, adaptation of parameters in this study is not coupled to perceived outcomes such as volatility in the environment.

see e.g. Diederer, K.M.J., and Schultz, W. (2015). Scaling prediction errors to reward variability benefits error-driven learning in humans. *J. Neurophysiol.* 114, 1628–1640.

The presented results detect a decrease of e.g. learning rate after migration which is in line with the literature where learning decreases as sampling variance decreases. This drop, however, should be steeper in low variance environments than in high variance environments.

A feature I could not detect in the results as they were presented. This could also explain the lack of difference in the predictions between the two variance conditions in Figure S7.

In an alternative approach (hierarchical Gaussian Fields), learning parameters depend on participants' experience of environmental volatility (besides other parameters).

These models have gained traction in computational neuroscience and have been applied also in social contexts.

Diaconescu, A.O., Mathys, C., Weber, L.A.E., Daunizeau, J., Kasper, L., Lomakina, E.I., Fehr, E., and Stephan, K.E. (2014). Inferring on the Intentions of Others by Hierarchical Bayesian Learning. *PLoS Comput Biol* 10, e1003810.

Mathys, C., Daunizeau, J., Friston, K.J., and Stephan, K.E. (2011). A Bayesian foundation for individual learning under uncertainty. *Front. Hum. Neurosci.* 5, 39.

A discussion on the above mentioned limitations could be helpful particularly as the authors want to use their models to explain cultural evolution and social learning that are based on evolved cognitive processes. Incorporating neurocomputational theories on cognitive processes could eventually inform theories of cultural evolution.

Minor points:

p.3/4 The authors state: "Despite its theoretical relevance, this difference in adaptive social learning strategies between spatially and temporally varying environments, has not been investigated empirically."

Given the literature cited above I think this is a bit of an overstatement and should be toned down.

I sign my reviews

Ulf Toelch, BIH QUEST Center for Transforming Biomedical Research

Decision letter (RSOS-200734.R0)

Dear Mr Deffner,

On behalf of the Editors, we are pleased to inform you that your Manuscript RSOS-200734 "Dynamic Social Learning in Temporally and Spatially Variable Environments" has been accepted for publication in Royal Society Open Science subject to minor revision in accordance with the referees' reports. Please find the referees' comments along with any feedback from the Editors below my signature.

Please submit your revised manuscript and required files (see below) no later than 7 days from today's (ie 10-Sep-2020) date. Note: the ScholarOne system will 'lock' if submission of the revision is attempted 7 or more days after the deadline. If you do not think you will be able to meet this deadline please contact the editorial office immediately.

on behalf of Dr Simone Schnall (Associate Editor) and Kevin Padian (Subject Editor)
openscience@royalsociety.org

Associate Editor Comments to Author (Dr Simone Schnall):

Dear Dr. Deffner,

Apologies for the delay and many thanks for your patience. I have received reviews from three highly knowledgeable experts, who were all very impressed with your work. The issues they raised are detailed in the reviews, so I will not reiterate them here.

I invite you to prepare a revision that follows their suggestions, and look forward to seeing the next version of the manuscript.

Sincerely,
Simone Schnall

Reviewer comments to Author:

Reviewer: 1
Comments to the Author(s)

See doc

Reviewer: 2
Comments to the Author(s)

This paper investigates social learning in temporally and spatially variable environments using lab experiments and computational learning models. The results show that individuals use both experience and frequency cues to direct their social information use. Interestingly, the results also show that participants selectively respond to different changes in the environment with more conformist social learning following spatial changes and more individual learning following

temporal changes. This latter result confirms previously untested theoretical predictions about how individuals should strategically use social information.

Overall, I found the paper of outstanding quality and I think the conclusions are an important contribution to the field of cultural evolution. The (preregistered) experiment is extremely well designed, the reporting of the data and methodology are very detailed and transparent to enable reproducing the results. Note that I cannot comment on computational learning models which are beyond the scope of my expertise. However, the strategy seems sound to me and, from the parts that I was able to follow, I find that the conclusions drawn from the results are valid. The paper is also extremely well written, which makes even the technical sections easy to read.

I have a couple of minor comments/questions, but overall, I think the paper can be published as is.

Given that experimental sessions “lasted between 30 and 60 minutes depending on the speed of participants”, I assume that participants did not face any time constraints to make their decisions. Under those conditions how to explain that participants only viewed 75-90% of boxes showing previous choices of other group members and 50-65% of boxes showing their respective levels of experience in the present region? In absence of costs associated with social information use, I would expect participants to reveal as much of social information as they can (especially because participants can only reveal up to 6 pieces of social information).

In the methods section, the Mouse Tracking subsection states that participants had to hover over the information boxes to see their “content”. I assume information was not visible anymore once participants moved the mouse elsewhere, is that correct? Could working memory limitations explain why participants did not tend to reveal the content of 100% of boxes?

In figure 2, y-axes range from 0.1 to 0.7 in A-B and 0.3 to 1 in C-D. This is slightly confusing, especially because the proportion of optimal choices in B seems to plateau. Distracted readers might incorrectly assume that it is because individuals reach 100% accuracy.

Reviewer: 3

Comments to the Author(s)

Defnner et al. describe in their manuscript "Dynamic Social Learning in Temporally and Spatially Variable Environments" a 4 armed bandit task where participant behaviour analysed via a computational model. The experiment and modelling is well planned and executed, particularly, the approach to first simulate data, then create the experiment, and later perform parameter recovery is something I would wish for in all computational modelling papers.

The methods are sound, but I have only briefly looked at the underlying code and have not performed an in depth review of it.

I have some comments the authors may want to consider.

1. Code of experimental software

Perhaps I missed it, but is it possible to make the oTree code available? I am not overly familiar with the software, but to allow replication of your experiment this would be helpful.

2. Adaptation of parameters.

The experimental environment includes often a high degree of uncertainty. Learning rates and other parameters are thus adjusted.

The authors adjust for this by a. modelling monotonic effects and also Gaussian processes.

This modelling step is underwritten. From the information in the Methods it was not possible for me to follow what the authors were doing but I had to resort to code and results sections. More detail is needed here. How are the actual parameters from Figure 4 related to the equation 2.7.

One example is the use of sigma in the monotonic models. In a first read through I thought this would refer to the balance between the social and individual information.

I also thought that this would be the only parameter that would be adjusted. Please add more information on the modelling process directly in the main manuscript.

3. Model limitations.

The adaptation of parameters is stochastic and not mechanistic/cognitive. That is, adaptation of parameters in this study is not coupled to perceived outcomes such as volatility in the environment.

see e.g. Diederer, K.M.J., and Schultz, W. (2015). Scaling prediction errors to reward variability benefits error-driven learning in humans. *J. Neurophysiol.* 114, 1628–1640.

The presented results detect a decrease of e.g. learning rate after migration which is in line with the literature where learning decreases as sampling variance decreases. This drop, however, should be steeper in low variance environments than in high variance environments.

A feature I could not detect in the results as they were presented. This could also explain the lack of difference in the predictions between the two variance conditions in Figure S7.

In an alternative approach (hierarchical Gaussian Fields), learning parameters depend on participants' experience of environmental volatility (besides other parameters).

These models have gained traction in computational neuroscience and have been applied also in social contexts.

Diaconescu, A.O., Mathys, C., Weber, L.A.E., Daunizeau, J., Kasper, L., Lomakina, E.I., Fehr, E., and Stephan, K.E. (2014). Inferring on the Intentions of Others by Hierarchical Bayesian Learning. *PLoS Comput Biol* 10, e1003810.

Mathys, C., Daunizeau, J., Friston, K.J., and Stephan, K.E. (2011). A Bayesian foundation for individual learning under uncertainty. *Front. Hum. Neurosci.* 5, 39.

A discussion on the above mentioned limitations could be helpful particularly as the authors want to use their models to explain cultural evolution and social learning that are based on evolved cognitive processes. Incorporating neurocomputational theories on cognitive processes could eventually inform theories of cultural evolution.

Minor points:

p.3/4 The authors state: "Despite its theoretical relevance, this difference in adaptive social learning strategies between spatially and temporally varying environments, has not been investigated empirically."

Given the literature cited above I think this is a bit of an overstatement and should be toned down.

I sign my reviews

Ulf Toelch, BIH QUEST Center for Transforming Biomedical Research

===PREPARING YOUR MANUSCRIPT===

===PREPARING YOUR REVISION IN SCHOLARONE===

<https://royalsociety.org/journals/authors/author-guidelines/#data>. You should ensure that you cite the dataset in your reference list. If you have deposited data etc in the Dryad repository,

please only include the 'For publication' link at this stage. You should remove the 'For review' link.

Author's Response to Decision Letter for (RSOS-200734.R0)

See Appendix B.

Decision letter (RSOS-200734.R1)

Dear Mr Deffner,

It is a pleasure to accept your manuscript entitled "Dynamic Social Learning in Temporally and Spatially Variable Environments" in its current form for publication in Royal Society Open Science. The comments from the Editors are included at the foot of this letter.

Best regards,
Lianne Parkhouse
Editorial Coordinator

on behalf of Dr Simone Schnall (Associate Editor) and Kevin Padian (Subject Editor)
openscience@royalsociety.org

Associate Editor Comments to Author (Dr Simone Schnall):

Dear Mr. Deffner,

Thanks for preparing such a thorough revision that responded to the reviewers' comments. It is my pleasure to accept the current version for publication. I believe your work makes a valuable addition to the literature, and therefore look forward to seeing the article in print in the journal.

Best wishes,
Simone Schnall

Appendix A

This paper presents a two group four-armed bandit laboratory experiment which included random migration and temporal changes in the optimal behaviour. Participants had access to the behaviour of their group mates and how long they had been in that group. Results showed that participants tended to favour conformist social learning following migration but favoured individual learning following temporal changes. Participants also biased learning towards those who had been in the group for a longer period of time.

The main findings make a solid contribution to the field of cultural evolution. The principle value of the paper however, is the exemplary use of quantitative methods applied to experimental data. The analysis is thorough, is laid out clearly and will help others adopt similar good practices in future research. The approach set out here can be used across a broad range of academic fields who have yet to adopt Bayesian approaches.

Suggestions / comments (bracketed numbers are line numbers)

- (88) The participants represent a WEIRD sample. I recommend the authors comment on whether the fit of the experimental results to cultural evolutionary theory suggests that the theory applies only to the participant sample, German-speakers, WEIRD people or humans in general.
- Relatedly, why do the authors think that the participants' behaviour was consistent with cultural evolutionary theory? Is it possible that cognitive development in WEIRD populations emphasises choice/goal/reward-based optimisation and thus may be particularly effective at resulting in behaviour consistent with theoretical predictions that are based on differential fitness?
- (94) 12 and 14 what? What currency is being used here?
- (146) It is not clear what "easy" and "hard" refer to in this case.
- (153) What was the result when payoff learning was used in the previous experiment?
Presumably, participants converged onto the most efficient choice.

- (154) I agree that direct access to payoff information is sometimes not available (certainly not as clear as experiments provide it), but it is probably also true that individuals rarely have no idea at all how their peers are doing. For example, relative crop yield may be easy to estimate. The wording currently implies that payoff learning was omitted just because it is preferred, essentially universally, to every other information source. If this is true, then the results are always conditional on assuming that payoff learning is not used. Perhaps the justification for omitting payoff learning could be just that the experiment did only seek to model these scenarios where the information is either completely absent or too unreliable to be useful.
- (164) Was the time spent viewing social information considered in any analysis? Could there be any link between viewing time and their chance to use the information / their eventual payoff? At least, it may be useful to have some idea, on average, how long participants spent viewing social information each round.
- (350) Does "all" here mean the entire posterior distribution or just all of particular region of it (e.g. 89% HDPI)
- (401) This sentence concerns future studies and so I recommend it goes in the Discussion section.
- (423) The results surrounding the experience bias were not completely clear to me. Does the experience bias arise instead of conformity or as well as? Further to this, was there any consideration of synergistic or contradictory information. For example, if individual A (with the longest tenure in the population) had chosen corn (the best option), but everyone else had chosen wheat, does one strategy overrides the other?
- (433) Did this study find any evidence that longer tenure individuals were copied less (in lieu of being out of date)? Please clarify.
- (440 & 457) The authors claim that the results 'confirm' theoretical expectations/predictions. I don't think this claim is valid. Instead, the results are 'consistent' (or equivalent) with theory.

- (459) If temporal changes in the best crop option were not announced, how would this change the conclusions described here for conformity? It might be assumed that conformity would reduce across the board as participants would need to continually experiment to investigate changes in their payoffs. Similarly, conformity arising from migration could result in participants becoming stuck on inefficient options even despite environmental changes. The results therefore are somewhat dependent on the assumption that information regarding temporal ecological changes is available. Admittedly, this assumption is probably valid in many circumstances, but this could warrant some brief discussion.
- (461) Cultural Group Selection is sprung on the reader in the Discussion without a description of what this is.

Appendix B

Associate Editor Comments to Author (Dr Simone Schnall):

Dear Dr. Deffner,

Apologies for the delay and many thanks for your patience. I have received reviews from three highly knowledgeable experts, who were all very impressed with your work. The issues they raised are detailed in the reviews, so I will not reiterate them here.

I invite you to prepare a revision that follows their suggestions, and look forward to seeing the next version of the manuscript.

Sincerely,
Simone Schnall

Dear Dr. Schnall,

Thank you very much. We totally understand the delay given the current situation and very much appreciate the time and effort that the reviewers and you have put into this manuscript. The comments have been very helpful in guiding us how best to improve and clarify our article.

Sincerely,

Dominik Deffner (unfortunately not yet Dr.)

Reviewer comments to Author:

Reviewer: 1

see doc

Comments to the Author(s)

This paper presents a two group four-armed bandit laboratory experiment which included random migration and temporal changes in the optimal behaviour. Participants had access to the behaviour of their group mates and how long they had been in that group. Results showed that participants tended to favour conformist social learning following migration but favoured individual learning following temporal changes. Participants also biased learning towards those who had been in the group for a longer period of time.

The main findings make a solid contribution to the field of cultural evolution. The principle value of the paper however, is the exemplary use of quantitative methods applied to experimental data. The analysis is thorough, is laid out clearly and will help others adopt similar good practices in future research. The approach set out here can be used across a broad range of academic fields who have yet to adopt Bayesian approaches.

Thank you very much!

Suggestions / comments (bracketed numbers are line numbers)

- (88) The participants represent a WEIRD sample. I recommend the authors comment on whether the fit of the experimental results to cultural evolutionary theory suggests that the theory applies only to the participant sample, German-speakers, WEIRD people or humans in general.

Thanks for raising this immensely important point. Generalizability beyond the study sample is always a concern for empirical studies and, unfortunately, we will not be able to resolve this issue in the present paper. While we do not know of any reason our results should apply to Germans in particular and we do not find the WEIRD/non-WEIRD dichotomy particularly useful, we totally agree that experimentalists should think harder about what populations they expect their results to generalize to. This does not only refer to different geographical locations but also to different social classes and milieus within societies. In this regard, we want to note that our sample does not constitute a typical student sample but is quite diverse in terms of age, education and employment. We now discuss the issues of generalizability and WEIRD populations in the discussion section:

“Studies in the behavioral and social sciences have been repeatedly criticized for relying almost exclusively on WEIRD samples [51]. Although not the typical student sample, participants in our study were also predominantly Western, relatively educated, and from an industrialized, rich, and democratic country. While we do not find the WEIRD/non-WEIRD dichotomy particularly useful, as it hides important cultural variation both within and between societies, generalizability of empirical findings to new environments, settings or populations is always a concern for experimental studies. This concern is typically addressed by (calls for) replication across different conditions and populations. Although replication across societies may suggest greater generalizability, it is impossible to sample all relevant populations and drawing inferences from either positive or negative results is difficult without a profound theoretical understanding of factors expected to cause cross-cultural differences and their distribution across populations. As a complement to this bottom-up, data-driven approach, researchers in computer science have put forward a rigorous formal framework for licensed transfers of causal effects from experimental studies to new populations, in which only observational studies can be conducted [52, 53]. Developing a similar framework for cross-cultural generalizability would be a major advance.” (lines 482 – 496)

- Relatedly, why do the authors think that the participants’ behaviour was consistent with cultural evolutionary theory? Is it possible that cognitive development in WEIRD populations emphasises choice/goal/reward-based optimisation and thus may be particularly effective at resulting in behaviour consistent with theoretical predictions that are based on differential fitness?
We do not claim that participants’ behavior was consistent with cultural evolutionary theory in general, but with a specific set of predictions derived from more domain-specific models (especially Nakahashi, Wakano & Henrich, 2012). Still, we would argue that choice/goal/reward-based optimization is not specific to WEIRD populations, but characteristic of all human populations, if not of living systems in general. It is true that specific goals such as optimizing monetary rewards are highly dependent on cultural institutions such as trade and markets, but humans across populations pursue rewards that are desirable to them (Nettle et al., 2013). In this experiment, we use money as a culture-appropriate incentive system, but the predictions hold as long as people are motivated to seek out any type of reward. More generally, being unstable, far-from-equilibrium systems, animals almost per definition try to predict and seek out states that are desirable to them in order to resist entropy and maintain their very existence (Schrodinger, 1944; Friston, 2010). This is also evidenced by the ubiquity of reinforcement learning across organisms, including mammals and birds (Domjan, 2014), larval *Drosophila* (Schleyer et al., 2015) and the sea slug *Aplysia* (Brembs et al., 2002).
- (94) 12 and 14 what? What currency is being used here?

Fixed. It was between 12€ and 14€.

- (146) It is not clear what “easy” and “hard” refer to in this case.
Thank you! We have further clarified the meaning of “easy” and “hard” phases now.
- (153) What was the result when payoff learning was used in the previous experiment? Presumably, participants converged onto the most efficient choice.
Yes, participants very rapidly converged on the optimal option. Thus, there was little additional information gained about learning strategies after a few rounds following environmental change.
- (154) I agree that direct access to payoff information is sometimes not available (certainly not as clear as experiments provide it), but it is probably also true that individuals rarely have no idea at all how their peers are doing. For example, relative crop yield may be easy to estimate. The wording currently implies that payoff learning was omitted just because it is preferred, essentially universally, to every other information source. If this is true, then the results are always conditional on assuming that payoff learning is not used. Perhaps the justification for omitting payoff learning could be just that the experiment did only seek to model these scenarios where the information is either completely absent or too unreliable to be useful.
Thanks for the suggestion! We have now changed the respective section: “We did not include payoff information from other participants, as it is unrealistic to assume that learners can reliably access other individuals’ payoffs in most real-world situations. Our experiment thus models only scenarios where this information is either absent or too unreliable.” (lines 152-155)
- (164) Was the time spent viewing social information considered in any analysis? Could there be any link between viewing time and their chance to use the information / their eventual payoff? At least, it may be useful to have some idea, on average, how long participants spent viewing social information each round.
Thanks for the suggestion. Our learning models conditioned only on whether participants viewed a given piece of information at all and not on the duration they spent looking at it. Analyzing detailed information search trajectories and their influence on choice would be a great avenue for future research as suggested in the text:

“Below, we use detailed mousetracking information to condition learning strategies on the sources of social information individuals actually accessed in each round. Analyzing full search strategies using for instance information foraging models [29] would be a fruitful avenue for future research.” (line 165-168)
- (350) Does "all" here mean the entire posterior distribution or just all of particular region of it (e.g. 89% HDPI)
Yes, as can be seen from Figure 3C (pink curve), virtually all posterior probability lay above 1 providing very strong evidence for conformist social learning after spatial changes.
- (401) This sentence concerns future studies and so I recommend it goes in the Discussion section.
Thanks. We have now combined this sentence with a section discussing related points raised by reviewer 3 and added the following section in the discussion:

“We used computational learning models to investigate how strategies varied among individuals and changed over time. Through post-hoc simulation, we found that simulated agents showed similar overall patterns of behavior but learned much slower than real human participants. Implementing more realistic individual learning models from computational neuroscience based on, for example, variational inference could be fruitful avenues for future research [46–48]. Additionally, the way our model estimates time-varying learning parameters through monotonic effects and Gaussian processes is purely stochastic, i.e. relying on statistical associations, and not mechanistic, i.e. grounded in cognitive processing. Alternatively, in hierarchical Gaussian field models, which have also been applied to social contexts, learning parameters can vary depending on the perceived reward variability of the environment potentially linking observed changes in learning strategies to underlying cognitive mechanisms [49, 50].” (lines 470 – 481)

- (423) The results surrounding the experience bias were not completely clear to me. Does the experience bias arise instead of conformity or as well as? Further to this, was there any consideration of synergistic or contradictory information. For example, if individual A (with the longest tenure in the population) had chosen corn (the best option), but everyone else had chosen wheat, does one strategy overrides the other?

Experience and conformist bias can operate at the same time each influencing the probability an actor chooses a specific option (see equations 2.4, 2.5 and 2.6). We estimate the relative reliance on both types on information (κ parameter) and the direction and strength of both strategies (f and β parameters). Therefore, individuals can rely on both frequency and experience information, one of them are both types of cues simultaneously. Often, both sources of information will point towards the same behavior or there is no variation in frequency and/or experience. In these cases, the model will learn nothing new about an individual’s strategy. In the case of contradictory information, that you mention, the model has greater opportunity to learn about and differentiate between both strategies, as they imply different choice probabilities.

- (433) Did this study find any evidence that longer tenure individuals were copied less (in lieu of being out of date)? Please clarify.

As there is no learning cost in this experiment and individuals could constantly update their behavior, more learning opportunities should generally be associated with more adaptive behavior and we did not expect individuals with longer tenure to be out of date. Therefore, we predicted that individuals should copy more experienced individuals throughout the experiment (see predictions in preregistration). This was also confirmed by our findings, as stated in the abstract: “After both types of change, they biased decisions towards more experienced group members”. While individuals were strongly biased towards more experienced individuals right after migration, over time, they became less likely to copy long tenure individuals. They never switched to a “copy the young” strategy, however.

- (440 & 457) The authors claim that the results ‘confirm’ theoretical expectations/predictions. I don’t think this claim is valid. Instead, the results are ‘consistent’ (or equivalent) with theory.

Thank you! We now only claim the results “support” or “corroborate” theoretical expectations.

- (459) If temporal changes in the best crop option were not announced, how would this change the conclusions described here for conformity? It might be assumed that conformity would reduce across the board as participants would need to continually experiment to investigate changes in their payoffs. Similarly, conformity arising from migration could result in participants becoming stuck on inefficient options even despite environmental changes. The results therefore are somewhat dependent on the

assumption that information regarding temporal ecological changes is available. Admittedly, this assumption is probably valid in many circumstances, but this could warrant some brief discussion. For both types of change, individuals were told that conditions might (or might not) change now, so that they still needed to learn from their payoffs whether the environment had actually changed or not. The below-chance behavior after temporal changes (Fig. 2B) suggests that individuals actually first continued with their previous choices until they noticed that payoffs have changed. We decided to give them a cue about a potential change in the environment to standardize the period following each type of change for analysis. Otherwise, individuals would have noticed environmental changes at different points in time (or not at all) generating additional non-systematic variation that is unrelated to the research questions we wanted to answer.

- (461) Cultural Group Selection is sprung on the reader in the Discussion without a description of what this is.

Thanks. We now added a sentence to further elaborate on this point: “This theory presupposes that alternative equilibria of local norms and behaviors are stabilized through learning processes in different groups and group-level selection between such culturally differentiated populations might then lead to the spread of cooperative social norms that benefit the cultural group.” (lines 466 – 469)

Reviewer: 2

Comments to the Author(s)

This paper investigates social learning in temporally and spatially variable environments using lab experiments and computational learning models. The results show that individuals use both experience and frequency cues to direct their social information use. Interestingly, the results also show that participants selectively respond to different changes in the environment with more conformist social learning following spatial changes and more individual learning following temporal changes. This latter result confirms previously untested theoretical predictions about how individuals should strategically use social information.

Overall, I found the paper of outstanding quality and I think the conclusions are an important contribution to the field of cultural evolution. The (preregistered) experiment is extremely well designed, the reporting of the data and methodology are very detailed and transparent to enable reproducing the results. Note that I cannot comment on computational learning models which are beyond the scope of my expertise. However, the strategy seems sound to me and, from the parts that I was able to follow, I find that the conclusions drawn from the results are valid. The paper is also extremely well written, which makes even the technical sections easy to read.

I have a couple of minor comments/questions, but overall, I think the paper can be published as is.

Thank you very much for the very encouraging feedback!

Given that experimental sessions “lasted between 30 and 60 minutes depending on the speed of participants”, I assume that participants did not face any time constraints to make their decisions. Under those conditions how to explain that participants only viewed 75-90% of boxes showing previous choices of other group members and 50-65% of boxes showing their respective levels of experience in the

present region? In absence of costs associated with social information use, I would expect participants to reveal as much of social information as they can (especially because participants can only reveal up to 6 pieces of social information).

Correct. As we wanted to constrain participants' behavior as little as possible to make the results more comparable to observational "real-world" settings, there was no particular time limitation. Therefore, rational actors should access all available sources of information. From the curves in Fig. 2, one can see that information use declined while proportions of correct choices increased over time. This seems to suggest that while participants became increasingly confident in their choices, they were looking less for social information. There was also considerable inter-individual variation in social information seeking. Some participants were accessing all information boxes throughout the experiment, while others at some point just started to choose an option without considering social information at all. This is in line with previous research that found evidence for strong and stable differences in social information use (e.g. Efferson et al., 2008).

In the methods section, the Mouse Tracking subsection states that participants had to hover over the information boxes to see their "content". I assume information was not visible anymore once participants moved the mouse elsewhere, is that correct? Could working memory limitations explain why participants did not tend to reveal the content of 100% of boxes?

Yes, this is correct. Information was only visible as long as the mouse was positioned over the box. As individuals could access each information box as often as they wanted, we do not think that working memory limitations can explain their lack of social information use. To the contrary, if they were not able to hold all pieces of information in working memory, we would expect them to open each box multiple times rather than not at all.

In figure 2, y-axes range from 0.1 to 0.7 in A-B and 0.3 to 1 in C-D. This is slightly confusing, especially because the proportion of optimal choices in B seems to plateau. Distracted readers might incorrectly assume that it is because individuals reach 100% accuracy.

Great point. This is fixed now. To avoid confusion about the y-axes, we now use the full range 0-1 for all four plots in Figure 2.

Reviewer: 3

Comments to the Author(s)

Deffner et al. describe in their manuscript "Dynamic Social Learning in Temporally and Spatially Variable Environments" a 4 armed bandit task where participant behaviour analysed via a computational model. The experiment and modelling is well planned and executed, particularly, the approach to first simulate data, then create the experiment, and later perform parameter recovery is something I would wish for in all computational modelling papers.

The methods are sound, but I have only briefly looked at the underlying code and have not performed an in depth review of it.

I have some comments the authors may want to consider.

1. Code of experimental software

Perhaps I missed it, but is it possible to make the oTree code available? I am not overly familiar with the

software, but to allow replication of your experiment this would be helpful.

Thanks for the great suggestion. We now included a folder with the full oTree experimental software in the accompanying GitHub repo (<https://github.com/DominikDeffner/Dynamic-Social-Learning>).

2. Adaptation of parameters.

The experimental environment includes often a high degree of uncertainty. Learning rates and other parameters are thus adjusted.

The authors adjust for this by a. modelling monotonic effects and also Gaussian processes. This modelling step is underwritten. From the information in the Methods it was not possible for me to follow what the authors were doing but I had to resort to code and results sections. More detail is needed here. How are the actual parameters from Figure 4 related to the equation 2.7. One example is the use of sigma in the monotonic models. In a first read through I thought this would refer to the balance between the social and individual information. I also thought that this would be the only parameter that would be adjusted. Please add more information on the modelling process directly in the main manuscript.

Thank you for the feedback. There is always a trade-off between providing too little information for particularly interested readers and overwhelming those possessing less statistical expertise with unnecessary technical detail. We added the following sentence to clarify that the same procedures apply to other learning parameters as well (lines 221-222): “Note we only describe sigma, the weight of social learning, in detail here, but other learning parameters were constructed in the same way.” We also added more detail on the Gaussian process approach and included a new equation for the multivariate normal prior that defines Gaussian processes (equation 2.8; lines 238 – 242).

3. Model limitations.

The adaptation of parameters is stochastic and not mechanistic/cognitive. That is, adaptation of parameters in this study is not coupled to perceived outcomes such as volatility in the environment. see e.g. Diederer, K.M.J., and Schultz, W. (2015). Scaling prediction errors to reward variability benefits error-driven learning in humans. *J. Neurophysiol.* 114, 1628–1640.

The presented results detect a decrease of e.g. learning rate after migration which is in line with the literature where learning decreases as sampling variance decreases. This drop, however, should be steeper in low variance environments than in high variance environments. A feature I could not detect in the results as they were presented. This could also explain the lack of difference in the predictions between the two variance conditions in Figure S7.

In an alternative approach (hierarchical Gaussian Fields), learning parameters depend on participants' experience of environmental volatility (besides other parameters).

These models have gained traction in computational neuroscience and have been applied also in social contexts.

Diaconescu, A.O., Mathys, C., Weber, L.A.E., Daunizeau, J., Kasper, L., Lomakina, E.I., Fehr, E., and Stephan, K.E. (2014). Inferring on the Intentions of Others by Hierarchical Bayesian Learning. *PLoS Comput Biol* 10, e1003810.

Mathys, C., Daunizeau, J., Friston, K.J., and Stephan, K.E. (2011). A Bayesian foundation for individual learning under uncertainty. *Front. Hum. Neurosci.* 5, 39.

A discussion on the above mentioned limitations could be helpful particularly as the authors want to use their models to explain cultural evolution and social learning that are based on evolved cognitive

processes. Incorporating neuro-computational theories on cognitive processes could eventually inform theories of cultural evolution.

This is a great point! We added a section in the discussion to discuss the potential relevance of more cognitively detailed learning models in cultural evolution:

“We used computational learning models to investigate how strategies varied among individuals and changed over time. Through post-hoc simulation, we found that simulated agents showed similar overall patterns of behavior but learned much slower than real human participants. Implementing more realistic individual learning models from computational neuroscience based on, for example, variational inference could be fruitful avenues for future research [46–48]. Additionally, the way our model estimates time-varying learning parameters through monotonic effects and Gaussian processes is purely stochastic, i.e. relying on statistical associations, and not mechanistic, i.e. grounded in cognitive processing. Alternatively, in hierarchical Gaussian field models, which have also been applied to social contexts, learning parameters can vary depending on the perceived reward variability of the environment potentially linking observed changes in learning strategies to underlying cognitive mechanisms [49, 50].” (lines 470 – 481)

Minor points:

p.3/4 The authors state: "Despite its theoretical relevance, this difference in adaptive social learning strategies between spatially and temporally varying environments, has not been investigated empirically." Given the literature cited above I think this is a bit of an overstatement and should be toned down.

Thank you. This is has been changed to (lines 55-57): “Despite its theoretical relevance, this difference in adaptive social learning strategies between spatially and temporally varying environments, has received little empirical attention.”

I sign my reviews

Ulf Toelch, BIH QUEST Center for Transforming Biomedical Research

Cited literature

- Brembs, B., Lorenzetti, F. D., Reyes, F. D., Baxter, D. A., & Byrne, J. H.** (2002). Operant reward learning in Aplysia: neuronal correlates and mechanisms. *Science*, 296(5573), 1706-1709.
- Domjan, M.** (2014). *The principles of learning and behavior*. Nelson Education.
- Efferson, C., Lalive, R., Richerson, P. J., McElreath, R., & Lubell, M.** (2008). Conformists and mavericks: the empirics of frequency-dependent cultural transmission. *Evolution and Human Behavior*, 29(1), 56-64.
- Friston, K.** (2010). The free-energy principle: a unified brain theory?. *Nature reviews neuroscience*, 11(2), 127-138.
- Nakahashi, W., Wakano, J. Y., & Henrich, J.** (2012). Adaptive social learning strategies in temporally and spatially varying environments. *Human Nature*, 23(4), 386-418.
- Nettle, D., Gibson, M. A., Lawson, D. W., & Sear, R.** (2013). Human behavioral ecology: current research and future prospects. *Behavioral Ecology*, 24(5), 1031-1040.
- Schleyer, M., Miura, D., Tanimura, T., & Gerber, B.** (2015). Learning the specific quality of taste reinforcement in larval Drosophila. *Elife*, 4, e04711.
- Schrodinger, E.** (1944). What is life? The physical aspect of the living cell. *Cambridge*.